SCIENCE FORUM

# Addressing selective reporting of experiments through predefined exclusion criteria

**Abstract** The pressure for every research article to tell a clear story often leads researchers in the life sciences to exclude experiments that 'did not work' when they write up their results. However, this practice can lead to reporting bias if the decisions about which experiments to exclude are taken after data have been collected and analyzed. Here we discuss how to balance clarity and thoroughness when reporting the results of research, and suggest that predefining the criteria for excluding experiments might help researchers to achieve this balance.

**KLEBER NEVES AND OLAVO B AMARAL\***

\*For correspondence: olavo@ bioqmed.ufrj.br

Competing interests: The authors declare that no competing interests exist.

## Introduction

Experiments fail all the time, for both complex and trivial reasons. Because of this, many experiments are repeated – typically after tinkering a bit with the protocol – and the initial failed attempts often go unreported in published articles. This is understandable: trying to present all the unsuccessful results of a long, obstacle-prone project might make an article almost unreadable. Scientific articles thus frequently privilege conciseness over completeness, in order to tell an intuitive story and facilitate the understanding of complex ideas (*Sanes, 2019*; *Sollaci and Pereira, 2004*).

Narrative quality, however, can go against the need for transparency in reporting to ensure reproducibility: as the process of conducting research is vulnerable to a large amount of biases (*Chavalarias and Ioannidis, 2010*), selective reporting of results can have detrimental consequences on the scientific record (*Nosek et al., 2015*; *Nissen et al., 2016*). If enough experiments are conducted, some are bound to attain significant results by chance alone using typical statistical standards (*Ioannidis, 2005*). Failure to fully report on all attempts to carry out an experiment, thus, can lead to a scenario where data can be cherry-picked to support one's hypothesis (*Simmons et al., 2011*).

As laboratory scientists, we understand both sides of the argument: reporting on every experimental failure will increase noise without adding much value to the reported results; on the other hand, having unlimited flexibility to decide whether an experiment can be excluded from a research article opens up a huge avenue for bias to creep in. Here we make the case that predefined inclusion and exclusion criteria for experiments can help solve this conundrum, and discuss ways in which they can be implemented in the workflow of experimental projects, particularly in those of a confirmatory nature. We also describe how we are taking this approach in the Brazilian Reproducibility Initiative, a large-scale multicenter replication of experimental findings in basic biomedical science (*Amaral et al., 2019*).

## The many-level file drawer

Selective reporting can appear at many levels. A considerable body of literature exists on the omission of whole studies, a phenomenon best known as the 'file drawer effect' (*Rosenthal, 1979*). This is best studied in areas such as clinical trials and psychology (*Dwan et al., 2013*; *Ferguson and Heene, 2012*; *Fanelli, 2012*), in

> # Confirmation bias can easily lead one to discard experiments that 'didn't work' by attributing the results to experimental artifacts.

which meta-analytic statistical methods are routinely used for estimating publication bias (*Jin et al., 2015*). At the level of analysis, there is also evidence of bias in selective reporting of measured outcomes within trials (*Williamson et al., 2005*). At the other end of the scale, at the level of data collection, there has been a reasonable amount of discussion about the selective post-hoc exclusion of data points identified as outliers (*Holman et al., 2016*).

Not reporting the results of individual experiments is an intermediate level of bias that lies between the omission of studies and the omission of data points. It appears to be common in scientific fields where a single article typically includes multiple experiments or datasets, as in much of the life sciences. Although this is potentially one of the largest sources of bias in bench research, it has been relatively underdiscussed. The case has been made that the prevalence of significant results within single articles is frequently too high to be credible, considering the statistical power of individual experiments (*Schimmack, 2012*; *Lakens and Etz, 2017*). However, such statistical evidence cannot identify whether this is due to experiments going missing, leading to selective reporting of positive results, or whether the published experiments are biased towards positive findings at the level of measurement or analysis. Whereas one can locate unpublished clinical trials because they are preregistered, or look for mismatching sample sizes in articles to infer removal/loss of subjects, detecting an unreported experiment requires information that is not usually available to the reader.

Once more, the problem is that reporting the full information on every experiment conducted within a project might be counterproductive as well. Laboratory science can be technically challenging, and experimental projects hardly ever run smoothly from start to finish; thus, a certain degree of selective reporting can be helpful to separate signal from noise. After all, hardly anyone would be interested to know that your histological sections failed to stain, or that your culture behaved in strange ways because of contamination.

What is the limit, however, to what can be left out of an article? While most scientists will agree that suppressing results from an article because they do not fit a hypothesis is unethical, few would argue in favor of including every methodological failure in it. However, if researchers are free to classify experiments into either category after the results are in, there will inevitably be room for bias in this decision.

## The reverse Texas sharpshooter

It is all too easy to find something that went wrong with a particular experiment to justify its exclusion. Maybe the controls looked different than before, or one remembers that someone had complained about that particular antibody vial. Maybe the animals seemed very stressed that day, or the student who ran the experiment didn't have a good hand for surgery. Or maybe an intentional protocol variation apparently made a previously observed effect disappear. This is particularly frequent in exploratory research, where protocols are typically adjusted along the way. It is thus common that people will repeat an experiment again and again with minor tweaks until a certain result is found – frequently one that confirms an intuition or a previous finding (e.g. 'I got it the first time, something must have gone wrong this time').

All of the factors above might be sensible reasons to exclude an experiment. However, this makes it all too easy to find a plausible explanation for a result that does not fit one's hypothesis. Confirmation bias can easily lead one to discard experiments that 'didn't work' by attributing the results to experimental artifacts. In this case, 'not working' conflates negative results – i.e. non-significant differences between groups – with methodological failures – i.e. an experiment that is uninterpretable because its outcome could not be adequately measured. Even in the best of intentions, a scientist with too much freedom to explore reasons to exclude an experiment will allow unconscious biases related to its results to influence his or her decision (*Holman et al., 2015*).

This problem is analogous to the forking paths in data analysis (*Gelman and Loken,*

**As it stands today, most basic research in the life sciences is performed in an exploratory manner, and should be taken as preliminary rather than confirmatory evidence.**

2013), or to the Texas sharpshooter fallacy (*Biemann, 2013*), in which hypothesizing after the results are known (HARKing) leads a claim to be artificially confirmed by the same data that inspired it (*Hollenbeck and Wright, 2017*). But while the Texas sharpshooter hits the bullseye because he or she draws the target at the point where the bullet landed, the scientist looking to invalidate an experiment draws his or her validation target away from the results – usually based on a problem that is only considered critical after the results are seen. Importantly, these decisions – and experiments – will be invisible in the final article if the norm is to set aside the pieces that do not fit the story.

### Paths to confirmatory research

One much-discussed solution to the problem of analysis flexibility is preregistration of hypotheses and methods (*Forstmeier et al., 2017*; *Nosek et al., 2018*). The practice is still mostly limited to areas such as clinical trials (in which registration is mandatory in many countries) and psychology (in which the movement has gained traction over reproducibility concerns), and is not easy to implement in laboratory science, where protocols are frequently decided over the course of a project, as hypotheses are built and remodeled along the way.

Although exploratory science is the backbone of most basic science projects, a confirmation step with preregistered methods could greatly improve the validation of published findings (*Mogil and Macleod, 2017*). As it stands today, most basic research in the life sciences is performed in an exploratory manner, and should be taken as preliminary rather than confirmatory evidence – more akin to a series of case reports than to a clinical trial. This type of research can still provide interesting and novel insights, but its weight as evidence for a given hypothesis

should be differentiated from that of confirmatory research that follows a predefined protocol (*Kimmelman et al., 2014*).

Interestingly, the concept of preregistration can also be applied to criteria that determine whether an experiment is methodologically sound or not, and thus amenable to suppression from a published article. Laboratory scientists are used to including controls to assess the internal validity of their methods. In PCR experiments, for instance, measures are typically taken along the way to alert the researcher when something goes wrong: the ratio of absorbance of an RNA sample at 280 and 260 nm is used as a purity test for the sample, and non-template controls are typically used to check for specificity of amplification (*Matlock, 2015*).

Rarely, however, are criteria for what constitutes an appropriate result for a positive or negative control decided and registered in advance, leaving the researcher free to make this decision once the results of the experiment are in. This not only fails to prevent bias, but actually adds degrees of freedom: much like adding variables or analysis options, adding methodological controls can provide the researcher with more possible justifications to exclude experiments (*Wicherts et al., 2016*). Once more, the solution seems to require that the criteria to discard an experiment based on these controls are set without seeing the results, in order to counter the possibility of bias.

Preregistration is not the only possible path to confirmatory research. Preregistering exclusion criteria may be unnecessary if the data for all experiments performed are presented with no censoring (*Oberauer and Lewandowsky, 2019*). In this setting, it is up to the reader to judge whether the data fit a hypothesis, as performing the entire set of possible analyses (*Steegen et al., 2016*) can show how much the conclusions depend on certain decisions, such as ignoring an experiment. However, this can only happen if all collected data are presented, which is not common practice in the life sciences (*Wallach et al., 2018*), partly because it goes against the tradition of conveying information in a narrative form (*Sanes, 2019*). If some form of data filtering – for failed experiments, noisy signals or uninformative data – is important for clarity (and this may very well be the case), preventing bias requires that exclusion criteria are set independently of the results.

A third option to address bias in the decision to include or exclude experiments is to perform blind data analysis – in which inclusion choices

are made by experts who are blinded to the results, but have solid background knowledge of the method in order to devise sensible criteria. This is commonly performed in physics, for instance, and allows validity criteria to be defined even after data collection (*MacCoun and Perlmutter, 2015*). Such a procedure might be less optimal than establishing and publically registering criteria, as preregistration offers additional advantages such as allowing unpublished studies to be tracked (*Powell-Smith and Goldacre, 2016*). Nevertheless, it is likely easier to implement, allows greater flexibility in analysis decisions, and can still go a long way in limiting selective reporting and analysis bias.

## Pre-specified criteria to clean up the data record

A solution to selective reporting, thus, is to set criteria to consider an experiment as valid, or a set of data as relevant for analysis, ideally before it is performed/collected. These include inclusion and exclusion criteria for animals or cultures to be used, positive and negative controls to determine if an assay is sensitive and/or specific, and additional variables or experiments to verify that an intervention's known effects have been observed. Ideally, these criteria should be as objective as possible, with thresholds and rules for when data must be included and when they should be discarded. They should also be independent of the outcome measure of the experiment – that is, the observed effect size should not be used as a basis for exclusion – and applied equally to all experimental groups. Importantly, when criteria for validity are met, this should be taken as evidence that the experiment is appropriate, and that it would thus be unethical to exclude it from an article reporting on the data.

As for any decision involving predefined thresholds, concerns over sensitivity and specificity arise: criteria that are too loose might lead to the inclusion of questionable or irrelevant data in an article, whereas those that are too stringent could lead meaningful experiments to be discarded. As with preregistration or statistical significance thresholds, this should not discourage researchers from addressing these limitations in an exploratory manner – one is always free to show data that does not fit validity criteria if this is clearly pointed out. What is important is that authors are transparent about it – and that the reader knows whether they are

following prespecified criteria to ignore an experiment or have decided on it after seeing the results. Importantly, this can only happen when data is shown – meaning that decisions to ignore an experiment with no predefined reason must inevitably be discussed alongside its results.

For widely used resources such as antibodies and cell lines, there are already a number of recommendations for validation that have been developed by large panels of experts and can be used for this purpose. For cell line authentication, for example, the International Cell Line Authentication Committee (ICLAC) recommends a $\geq$ 80% match threshold for short-tandem-repeat (STR) profiling, which allows for some variation between passages (e.g., due to genetic drift; *Capes-Davis et al., 2013*). For antibodies, the International Working Group for Antibody Validation recommends a set of strategies for validation (*Uhlen et al., 2016*), such as quantitative immunoprecipitation assays that use predefined thresholds for antibody specificity (*Marcon et al., 2015*). Other areas and methods are likely to have similarly established guidelines that can be used as references for setting inclusion and exclusion criteria for experiments.

As coordinators of the Brazilian Reproducibility Initiative, a multicenter replication of 60–100 experiments from the Brazilian biomedical literature over the last 20 years, conducted by a team of more than 60 labs, we have been faced with the need for validation criteria in many stages during protocol development (*Amaral et al., 2019*). As the project is meant to be confirmatory in nature, we intend to preregister every protocol, including the analysis plan. Furthermore, to make sure that each replication is methodologically sound, we are encouraging laboratories to add as many additional controls as they judge necessary to each experiment. To deal with the problem raised in this essay, however, we also require that they prespecify their criteria for using the data from these controls in the analysis.

For RT-PCR experiments, for instance, controls for RNA integrity and purity must be accompanied by which ratios will allow inclusion of the sample in the final experiment – or, conversely, will lead data to be discarded. For cell viability experiments using the MTT assay, positive controls for cell toxicity are recommended to test the sensitivity of the assay, but must include thresholds for inclusion of the experiment (e.g., a reduction of at least X% in cell viability). For behavioral experiments, accessory

# Agreement with predictions or previous findings should never be criteria for including an experiment in an article.

measurements to evaluate an intervention's known effects (such as weight in the case of high-calorie diets) can be used to confirm that it has worked as expected, and that testing its effects on other variables is warranted. Once more, thresholds must be set beforehand, and failure to meet inclusion criteria will lead the experiment to be considered invalid and repeated in order to attain a usable result.

Defining validation criteria in advance for every experiment has not been an easy exercise: even though researchers routinely devise controls for their experiments, they are not used to setting objective criteria to decide whether or not the results of an experiment should be included when reporting the data. However, in a preregistered, confirmatory project, we feel that this is vital to allow us to decide if a failure to replicate a result represents a contradiction of the original finding or is due to a methodological artifact. Moreover, we feel that predefining validation criteria will help to protect the project from criticism about a lack of technical expertise by the replicating labs, which has been a common response to failed replication attempts in other fields (*Baumeister, 2019*).

As one cannot anticipate all possible problems, it is likely that, in some experiments at least, such prespecified criteria might turn out not to be ideal in separating successful experiments from failed ones. Nevertheless, for the sake of transparency, we feel that it is important that any post-hoc decisions for considering experiments as unreliable are marked as such, and that both the decision and its impact on the results are disclosed. Once more, there is no inherent problem with exploratory research or data-dependent choices; the problem is when these are done secretly and communicated selectively (*Hollenbeck and Wright, 2017*; *Simmons et al., 2011*).

## Conclusion

Although we have focused on the use of validation criteria to make decisions about experiments, they can also be used to make decisions about which data to analyze. In fields such as electrophysiology or functional neuroimaging, for example, data typically pass through preprocessing pipelines before analysis: the use of predefined validation criteria could thus prevent the introduction of bias by researchers when exploring these pipelines (*Phillips, 2004*; *Carp, 2012*; *Botvinik-Nezer et al., 2019*). Genomics and a number of other high-throughput fields have also developed standard evaluation criteria to avoid bias in analysis (*Kang et al., 2012*). This suggests that communities centering on specific methods can reach a consensus on which criteria are minimally necessary to draw the line between data that can be censored and those that must be analyzed.

Such changes will only happen on a larger scale, however, if researchers are aware of the potential impacts of post-hoc exclusion decisions on the reliability of results, an area in which the life sciences still lag behind other fields of research. Meanwhile, individual researchers focusing on transparency and reproducibility should consider the possibility of setting – and ideally registering – predefined inclusion and exclusion criteria for experiments in their protocols. Some recommendations to consider include the following:

- whenever possible, prespecify and register the criteria that will be used to define whether an experiment is valid for analysis – or, conversely, whether it should be excluded from it;
- do not use criteria based on the effect size of the outcome measures of interest: agreement with predictions or previous findings should never be criteria for including an experiment in an article.
- implement public preregistration (*Nosek et al., 2018*), blind analysis (*MacCoun and Perlmutter, 2015*) and/or full data reporting with multiverse analysis (*Steegen et al., 2016*) to ensure that data inclusion choices are transparent and independent of the data.

Making these criteria as objective as possible can help researchers make inclusion decisions in an unbiased way, avoiding reliance on 'gut feelings' that can easily lead one astray. As Richard Feynman once said: 'science is a way of trying not to fool yourself, and you are the easiest person to fool' (*Feynman, 1974*). An easy way to

make this advice heard is to explicitly state what an appropriate experiment means before starting it, and to stick to your view after the results are in.

**Kleber Neves** is in the Institute of Medical Biochemistry Leopoldo de Meis, Federal University of Rio de Janeiro, Rio de Janeiro, Brazil

https://orcid.org/0000-0001-9519-4909

**Olavo B Amaral** is in the Institute of Medical Biochemistry Leopoldo de Meis, Federal University of Rio de Janeiro, Rio de Janeiro, Brazil

olavo@bioqmed.ufrj.br

https://orcid.org/0000-0002-4299-8978

_Author contributions:_ Kleber Neves, Writing - original draft; Olavo B Amaral, Conceptualization, Writing - review and editing

_Competing interests:_ The authors declare that no competing interests exist.

_Received_ 05 March 2020
_Accepted_ 15 May 2020
_Published_ 22 May 2020

### Funding

| Funder | Grant reference number | Author |
| --- | --- | --- |
| Serrapilheira Institute | Brazilian Reproducibility Initiative | Kleber Neves |

The funders had no role in study design, data collection and interpretation, or the decision to submit the work for publication.

**Decision letter and Author response**
Decision letter https://doi.org/10.7554/eLife.56626.sa1
Author response https://doi.org/10.7554/eLife.56626.sa2

## Additional files

### Data availability

There are no data associated with this article.

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
