## [Decision Letter]

Thank you for submitting your article "Addressing selective reporting of experiments – the case for predefined exclusion criteria" for consideration by *eLife*. Your article has been reviewed by two peer reviewers, both of whom have agreed to reveal their identity: Anita E Bandrowski (Reviewer #1); Wolfgang Forstmeier (Reviewer #2). If you are able to address the points raised by the reviewers (see below) in a revised version, we will be happy to accept your article for publication.

Summary:

Reviewer #1

This is an excellent article about a very important and largely overlooked topic in the reproducibility conversation. It absolutely should be published. As an experimental scientist, I appreciate the author's practical stance, something that many conversations miss. Indeed, in that vein I am suggesting a few tidbits that may help the article become a better reference for experimental scientists, not just a very interesting position paper. However, these are merely suggestions and the manuscript should be published with or without addressing them.

Reviewer #2

I think this manuscript makes a very useful contribution by promoting higher standards of scientific objectivity. The issue of selective reporting of experiments within publications that present several experiments has received only limited attention so far. Hence, the present paper is helpful in terms of inspiring a debate about scientific objectivity and in terms of clarifying to some extent what is and what is not legitimate. I think the present manuscript could still be improved in two ways.

Essential revisions:

1) The authors state: "...only way to prevent bias is by setting exclusion criteria beforehand. "

This is a very interesting argument, but I believe that there are other possibilities. Preregistration has not eliminated bias from the psychology literature, instead there are now many abandoned preregistrations. While this is a better state than biomedicine, where we really don't know who attempted which study, it is still not satisfying.

One could argue that it may become possible to label the study as exploratory vs validated. It seems that failure to validate is mostly problematic when clinical trials are involved and those might be reserved for only studies that have been validated using preregistration, proper controls, power analysis, blinding and a full description of the experimental group subjects. These criteria are addressed in less than 10% of studies, in our experience (MacLeod's laboratory has looked at percentages of rigor criteria across certain aspects of the disease literature much earlier than we did).

It should be noted, however, that exploratory studies can provide glimpses of interesting results not easily ascertained by holding all variables constant. One might argue that the real problem is that these exploratory studies are treated like real validated studies instead of something more akin to case reports. There is a good reason to have case reports, they are intended to describe an interesting observation. If biomedicine adopts a similar labeling system, it may also help to put the study in context.

2. In several sections, the authors describe antibody and cell lines examples, but give more practical examples of validation criteria for RT-PCR.

What should help the paper is a bit of practical advice for studies involving antibodies and cell lines. Across the biomedical literature, there are some technique specific advice that has been offered, not by a single expert, but by large panels of experts that have met and created a set of recommendations for properly validating the techniques.

Please consider adding some resources for different experimental paradigms in the form of a set of key references.

For antibodies I would suggest the Ulhen et al., 2016: PMID: 27595404 paper (based on a panel of experts).

For cell lines, I would suggest ICLAC.org (again a panel of experts with specific guidelines; there are multiple papers here, but probably pointing to the organization is sufficient).

3. I think it would be good if the paper could end with some clear messages (recommendations presented as numbered bullet points), which every study (i.e. publication) should consider. I would like to leave this up to the authors, but it should contain the message that the decision about the validity of an experiment must not be based on the effect size of the outcome measure.

4. I think that the paper would benefit from widening its scope such that its messages are applicable not only to preregistered studies but also to others. Rather than saying that scientific objectivity would require preregistration of criteria of experimental validity, I would say that there are 3 ways of achieving high objectivity standards: i) Preregistered criteria of validity; ii) Blinding of the person, who develops and applies the criteria of validity, from the research outcome of each experiment; iii) Complete reporting and summary of all experiments and pilot trials.

Currently the paper already hints at the possibility iii) in the subsection “Preregistration in confirmatory research” as an alternative to preregistration. The idea of blinding the decision maker has been explained elsewhere (eg MacCoun & Perlmutter, 2015; MacCoun & Perlmutter, 2017.

I guess it should be feasible for many studies to find an independent expert who is blinded from the data (outcome of experiments) and who can develop criteria for validity of experiments (even after data collection) and who applies those criteria strictly without knowing how this affects the overall outcome and conclusions. If studies implement such blinding techniques, I think it is also helpful to see these procedures described in the methods section. This signals awareness and will increase the credibility of a study compared to other studies where the authors may not even be aware of the issue of confirmation bias and selective reporting.

- MacCoun, R., & Perlmutter, S. (2015). Blind analysis: Hide results to seek the truth. Nature, 526(7572), 187-189.

- MacCoun, R. J., & Perlmutter, S. (2017). Blind analysis as a correction for confirmatory bias in physics and in psychology. Psychological science under scrutiny: Recent challenges and proposed solutions, 297-322.

---

## [Author Response]

[We repeat the reviewers’ points here in italic, and include our replies point by point, as well as a description of the changes made, in plain text].

Essential revisions:1) The authors state: "...only way to prevent bias is by setting exclusion criteria beforehand. "This is a very interesting argument, but I believe that there are other possibilities. Preregistration has not eliminated bias from the psychology literature, instead there are now many abandoned preregistrations. While this is a better state than biomedicine, where we really don't know who attempted which study, it is still not satisfying.

We agree with the reviewers that preregistrations are neither the only solution nor an infallible one. To address the reviewer’s concern, we have changed the passage mentioned to avoid conveying that interpretation. That said, we believe the point of preregistrations is to have a track record of what was originally planned, even if this is later abandoned and/or changed (as discussed in the subsection “Paths to confirmatory research”). Thus, the fact that not every preregistered study is published and not every preregistered protocol is followed faithfully is not necessarily an argument against the practice – one could argue that preregistration will at least allow a reader to know what remains unpublished or was changed in these cases. In any case, conforming to the reviewers’ suggestions, we now also discuss alternatives to preregistration to address reporting bias (see subsection “Paths to confirmatory research” and response to point #4 below).

One could argue that it may become possible to label the study as exploratory vs validated. It seems that failure to validate is mostly problematic when clinical trials are involved and those might be reserved for only studies that have been validated using preregistration, proper controls, power analysis, blinding and a full description of the experimental group subjects. These criteria are addressed in less than 10% of studies, in our experience (MacLeod's laboratory has looked at percentages of rigor criteria across certain aspects of the disease literature much earlier than we did).It should be noted, however, that exploratory studies can provide glimpses of interesting results not easily ascertained by holding all variables constant. One might argue that the real problem is that these exploratory studies are treated like real validated studies instead of something more akin to case reports. There is a good reason to have case reports, they are intended to describe an interesting observation. If biomedicine adopts a similar labeling system, it may also help to put the study in context.

We agree with the reviewer that most of basic biomedical research is indeed analogous to case reports or small exploratory studies in clinical research, and that a small fraction of basic science studies can be thought of as confirmatory research in the same sense as a large clinical trial. We also agree that it’s important to point out that this does not make them worthless – in fact, it can be argued that most of discovery science should be exploratory in nature –, and that a way to explicitly label studies as exploratory or confirmatory could be of use. We now highlight this distinction more clearly (using the case report analysis suggested) in the subsection “The reverse Texas sharpshooter”.

2. In several sections, the authors describe antibody and cell lines examples, but give more practical examples of validation criteria for RT-PCR.What should help the paper is a bit of practical advice for studies involving antibodies and cell lines. Across the biomedical literature, there are some technique specific advice that has been offered, not by a single expert, but by large panels of experts that have met and created a set of recommendations for properly validating the techniques.Please consider adding some resources for different experimental paradigms in the form of a set of key references.For antibodies I would suggest the Ulhen et al., 2016: PMID: 27595404 paper (based on a panel of experts).For cell lines, I would suggest ICLAC.org (again a panel of experts with specific guidelines; there are multiple papers here, but probably pointing to the organization is sufficient).

We thank the reviewer for pointing out these excellent references. We had indeed overemphasized certain methods in our original manuscript due to our experience in the Brazilian Reproducibility Initiative – which drove us to write this piece and is currently limited to three methods (MTT assays, RT-PCR and elevated plus maze), although it might add antibody-based techniques (Western blotting and/or immunohistochemistry) in the future.

We have used the references provided to add a paragraph about the use of pre-specified criteria for validation in cell line authentication and antibody quality control (see subsection “Pre-specified criteria to clean up the data record”), as suggested. We are aware that this list is still far from extensive, but such examples are indeed useful to drive home the kind of criteria that we are advocating for.

3. I think it would be good if the paper could end with some clear messages (recommendations presented as numbered bullet points), which every study (i.e. publication) should consider. I would like to leave this up to the authors, but it should contain the message that the decision about the validity of an experiment must not be based on the effect size of the outcome measure.

We agree with the reviewer that clear recommendations are useful, and have tried to do this by adding bullet points to the Conclusion section as suggested, making an effort to be very brief and straight to the point. We also fully agree with the message that the validity of an experiment cannot be based on the effect size of outcome measure, and now state this explicitly in subsection “Pre-specified criteria to clean up the data record” and Conclusion section.

4. I think that the paper would benefit from widening its scope such that its messages are applicable not only to preregistered studies but also to others. Rather than saying that scientific objectivity would require preregistration of criteria of experimental validity, I would say that there are 3 ways of achieving high objectivity standards: i) Preregistered criteria of validity; ii) Blinding of the person, who develops and applies the criteria of validity, from the research outcome of each experiment; iii) Complete reporting and summary of all experiments and pilot trials.Currently the paper already hints at the possibility iii) in the subsection “Preregistration in confirmatory research” as an alternative to preregistration. The idea of blinding the decision maker has been explained elsewhere (eg MacCoun & Perlmutter, 2015; MacCoun & Perlmutter, 2017.I guess it should be feasible for many studies to find an independent expert who is blinded from the data (outcome of experiments) and who can develop criteria for validity of experiments (even after data collection) and who applies those criteria strictly without knowing how this affects the overall outcome and conclusions. If studies implement such blinding techniques, I think it is also helpful to see these procedures described in the methods section. This signals awareness and will increase the credibility of a study compared to other studies where the authors may not even be aware of the issue of confirmation bias and selective reporting.- MacCoun, R., & Perlmutter, S. (2015). Blind analysis: Hide results to seek the truth. Nature, 526(7572), 187-189.- MacCoun, R. J., & Perlmutter, S. (2017). Blind analysis as a correction for confirmatory bias in physics and in psychology. Psychological science under scrutiny: Recent challenges and proposed solutions, 297-322.

We agree with the reviewer that there are options to preregistration in order for research to increase its confirmatory value, including independent blind analysis. We have thus widened the scope of the section on preregistration (now titled “Paths to confirmatory research”), which now describes different ways in which research may move towards confirmatory status, and also mention blind analysis as an alternative in the Conclusion section. That said, we do believe that preregistration offers some additional advantages to blind analysis, in the sense of making the record of planning public and helping to address publication bias as well. This is now discussed in the subsection “Pre-specified criteria to clean up the data record”.